# Entomopathogenic Fungi Infecting Lepidopteran Larvae: A Case from Central Argentina

**DOI:** 10.3390/life12070974

**Published:** 2022-06-29

**Authors:** Robin Gielen, Gerardo Robledo, Adriana Inés Zapata, Toomas Tammaru, Kadri Põldmaa

**Affiliations:** 1Department of Zoology, Institute of Ecology and Earth Sciences, Faculty of Science and Technology, University of Tartu, 50409 Tartu, Estonia; toomas.tammaru@ut.ee; 2BioTECA3, Centro de Biotecnología Aplicada al Agro y Alimentos, Facultad de Ciencias Agropecuarias, Universidad Nacional de Córdoba, Córdoba 5016, Argentina; gerardo.robledo@argo.unc.edu.ar; 3CONICET, Consejo Nacional de Investigaciones Cientificas y Técnicas, Godoy Cruz 2290, PC C1425FQB, Buenos Aires 1425, Argentina; 4Facultad de Ciencias Exactas, Físicas y Naturales, Universidad Nacional de Córdoba, Córdoba 5016, Argentina; adrzapata@unc.edu.ar; 5Department of Botany, Institute of Ecology and Earth Sciences, Faculty of Science and Technology, University of Tartu, 50409 Tartu, Estonia; hypomyc@ut.ee; 6Natural History Museum and Botanical Garden, University of Tartu, 51003 Tartu, Estonia

**Keywords:** hypocreales, noctuidae, host plant use, mortality, *Fusarium* *fujikuroi* and *Fusarium* *solani* species complex

## Abstract

Immature stages of insects are vulnerable to various antagonists, including pathogens. While the abiotic factors affecting pathogen prevalence in insect populations are reasonably well documented, much less is known about relevant ecological interactions. We studied the probability of the larvae of three lepidopteran species to die from fungal infection as a function of insect species and food plants in central Argentina. Local free-growing food plants were used to feed the lepidopteran larvae. The prevalence of entomopathogenic fungi remained low (about 5%), which is a value well consistent with observations on similar systems in other regions. Eight fungal species recorded, primarily belonging to *Fusarium* and *Aspergillus*, add evidence to the reconsideration of the nutritional modes in these genera in distinguishing the role of some species (complexes) to cause insect infections. Food plant species were found to have a substantial effect on the prevalence of entomopathogenic fungi. This was especially clear for the most abundant fungal species, a representative of the *Fusarium* *fujikuroi* complex. Feeding on a particular plant taxon can thus have a specific fitness cost. Compared to the data collected from Northern Europe, the Argentinian assemblages from the families Aspergillaceae and Nectriaceae overlapped at the genus level but did not share species. It remains to be confirmed if this level of divergence in the composition of assemblages of entomopathogenic fungi among distant regions represents a global pattern.

## 1. Introduction

Entomopathogenic fungi are present in perhaps all insect populations and thus have a potential for major ecological impact [1,2]. The latter appears to be insufficiently known, however. Besides taxonomic studies [3,4], research on entomopathogenic fungi has mostly been focused on their physiological host range [1,5]. The physiological host range of an entomopathogen, as defined by Hajek and Goettel [6], reflects its infectivity in laboratory experiments, which usually overestimates the host range in natural communities (ecological host range) [6,7]. The ecological host range must be constrained by the effects of various environmental factors, ranging from abiotic ones (temperature, humidity, UV) to various ecological interactions (host availability, host condition, host food plant, etc.) [2]. Of these, climatic factors have received the most attention, but various aspects of the host–parasite interaction in natural settings remain poorly understood. As one particular aspect, we know very little about the selective pressures entomopathogenic fungi exert on life histories of their hosts [8,9]. Even more fundamentally, it appears that documenting the prevalence of fungal pathogens is of considerable value, per se. Indeed, in contrast to a respectable amount of studies on larval mortality through predation or bacterial and viral pathogens [10], respective data on entomopathogenic fungi are notably scarce.

The knowledge on the role of the food plant as the mediator of relationships between herbivorous insects and its fungal pathogens is accumulating. For a long time, it has been suggested that food plants of herbivorous insects might provide a significant habitat for antagonists of insects, in accordance with the so called “Bodyguard hypothesis” [11,12]. This view is backed by recent discoveries of entomopathogenic fungi being able to live in plants as endophytes [13,14,15], in which the plant provides a stable environment for fungi, with the fungus offering protection against herbivores in return. Plants may not only provide favorable conditions in their tissues but also on their surface, creating suitable microclimatic conditions (protection from UV, higher relative humidity) for fungal spores [16]. If plants harbor significant amounts of entomopathogens, the differential probability of gaining fungal infection from different plant species can impose a selective pressure on insects’ food plant use [8].

From the perspective of the biology of fungi, an open question is where to draw the line between entomopathogenic and saprotrophic species [7]. Data are accumulating to indicate that some members of the genera *Fusarium* Link, *Aspergillus* P. Micheli and *Penicillium* Link, previously considered to comprise mostly saprotrophs but also plant pathogens, actually have the ability to infect and kill living insects [5,13,15,17]. This may have gone unnoticed because earlier ecological studies (at least on Lepidoptera) have primarily focused on fungal infections of the pupal and the adult stage, disregarding larval mortality. This may cause a substantial bias, as it has been shown that fungal communities infecting insects differ between life stages of the host [9], and further studies focusing on the larval stage are definitely warranted. 

To contribute to filling the outlined gaps, we performed an experiment in which we reared larvae of three species of Lepidoptera in a laboratory setting. The larvae were fed with field collected food plants representing five species. We recorded the prevalence of entomopathogenic fungi and compared it among insect and food plant species. Performed in central Argentina, the present study provides—when being contrasted to similar studies from Europe—an opportunity to compare species composition of entomopathogens on different continents and in different biomes, and also provides further evidence on the entomopathogenicity of fungi from the families Aspergillaceae Link and Nectriaceae Tul. and C. Tul.

## 2. Materials and Methods

To record the influence of biotic factors on the prevalence of entomopathogenic fungi in larvae, three polyphagous species of Lepidoptera—*Anicla infecta* (Ochsenheimer) (Noctuidae: Noctuinae), *Dargida albilinea* (Hübner) (Noctuidae: Hadeninae), and *Hypercompe indecisa* Walker (Erebidae: Arctiinae)—were reared at Instituto Multidisciplinario de Biologia Vegetal (IMBIV), CONICET—Universidad Nacional de Córdoba, in Córdoba, Argentina. Adult female moths were collected using light trapping in summer 2019–2020 (Table 1). Captured adults were placed individually into 100 cm^3^ cardboard boxes for oviposition. Newly hatched larvae were transferred individually to sterile 50 mL plastic vials with pierced lids and held indoors; temperatures were ranging from 24 to 30 °C. Each larva was assigned to one of the five treatments (food plant species). The larvae were reared until pupation on one particular plant species (frequently a natural situation for Lepidoptera, [18,19]), and checked every third day (daily in case of last instar) for survival and visual signs of fungal infection. The fate of the pupae could not be followed. The design of the experiment closely follows a similar study performed in Europe [9].

Five plant species common in the area were included into the study as food to the larvae: *Prosopis alba* Griseb. (Fabaceae), *Erigeron bonariensis* L. (Asteraceae), *E. sumatrensis* Retz. (Asteraceae), *Pascalia glauca* Ortega (Asteraceae), and *Zea mays* L. (Poaceae). Plant leaves the larvae were fed on were collected from IMBIV garden, Ciudad Universitaria (31°26′02″ S, 64°11′35″ W), which is a park inside Córdoba city where plants of native species grow. No chemicals (pesticides or other) had been applied to the plants. Three individual *Prosopis alba* trees were involved. Each larva was fed with the leaves of one tree individual throughout its development. A similar approach could not have been applied in the case of other plant species, as a single individual would not have had sufficient amount of biomass. The food was renewed every third day during the inspection of the larvae.

To identify the fungi and preserve them as pure living cultures, we sampled visible fungal material (only anamorphs were encountered) and inoculated these into Petri dishes with 2% malt extract agar (Oxoid, Cambridge, UK). A culture isolate representing each morphotype was subjected to DNA extraction. The procedures of growing the mycelium, extracting DNA, conducting PCR, and sequencing followed the protocols described by Põldmaa et al. [20]. Sequencing was performed at Macrogen Europe BV (Amsterdam, The Netherlands), using the available Standard-Seq service. The sequences along with their metadata were uploaded in PlutoF, a data management and publishing platform [21], and made available via the UNITE database [22]. The UNITE species hypotheses (SH) served as the basis for species identification by choosing an appropriate distance threshold value [23] in each case (Table 2). The advantage of the SH system is that regardless of the change of a Latin binomial, unique persistent identifiers, assigned to all SHs in the form of DOIs, allow for unambiguous communication about the identity of studied organisms. 

## 3. Results

Fungal infections were found on 37 (≈3.8%) of the 978 reared larvae. Fungi were detected on 2.9%, 2.8%, and 4.5% of the larvae of *A. infecta* (7/239), *D. albilinea* (6/209), and *H. indecisa* (24/530), respectively. All in all, ribosomal DNA full ITS and partial LSU sequences were obtained from 23 fungal isolates. In total, eight species-level taxa, belonging to three families of fungi, were detected on larvae which died in the course of rearing (Table 2). The most abundant genera were *Fusarium* and *Aspergillus*, which encompassed 67.5% and 21.6% of all infected specimens, respectively. All individual insect–fungus–host plant records and representative images of voucher material can be retrieved from https://dx.doi.org/10.15156/BIO/2483906.

When studying the dependence of the incidence of fungal infections on host insect and food plant species, the latter factor attained statistical significance (Table 3, Figure 1). The prevalence of fungal pathogens was 7.1% on both *E. bonariensis* and *Z. mays* (13/182 and 11/155, respectively), 2.4% on *E. sumatrensis* (3/125), 2.1% on *P. alba* (8/382), and 1.5% on *P. glauca* (2/134).

When analyzing the prevalence of the most abundant fungal genus (*Fusarium*) separately (Table 3), neither factor (host insect or food plant) attained statistical significance as a predictor of fungal incidence. However, for the most abundant species, a *Fusarium* species from the *F. fujikuroi* complex, its higher prevalence on the larvae fed with *E. bonariensis* was clear (Table 3).

## 4. Discussion

The prevalence of entomopathogenic fungi in larvae of three lepidopteran species reared in central Argentina was found to be low, remaining under 5%. This is in good quantitative concordance with the authors’ previous findings in similar studies in the north European forest zone [9,25]. Interestingly, fungal prevalence did not differ between lepidopteran species, while some—though not dramatic—among-species differences were detected in the Estonian study [9]. Even if we consider that the present study was conducted in laboratory conditions (though using freely growing host plants) and may not quantitatively reflect the situation in the field, we see our results as corroborating the emerging picture that entomopathogenic fungi are always present in insect populations and/or rearings, but do not usually cause epidemic outbreaks. With notable consistency, comparable studies have revealed prevalences in the magnitude of a few percent points.

Evidence has been accumulating in support of a facultatively endophytic lifestyle in entomopathogenic fungi, with—*Beauveria bassiana* (Bals.-Criv.) Vuill. [14], *Akanthomyces muscarius* (Petch) Spatafora, Kepler and B. Shrestha [4,15], and *Metarhizium anisopliae* (Metschn.) Sorokīn [26,27] providing just some examples. Though some dispersal of the fungal propagules through the air cannot be excluded [2], we believe that, in our experiment, the insects became infected by the mediation of their free-growing food plants. This assumption is supported by the clear effect of host plant species on the prevalence of fungal infections, detected in this study (Table 3, Figure 1). The effect of the plant was statistically significant in the whole dataset and also in the most abundant fungal species, a representative of the *Fusarium fujikuroi* complex. Moreover, all the involved fungal genera have been reported to occur as endophytes in a study from Brazil [13].

Our sample does not allow for a meaningful analysis of the association between particular species of plants and fungi. Nevertheless, interestingly, none of the fungal taxa recorded more than twice were found infecting larvae fed with plants from just one family. This pattern is in some conflict with the idea of a high degree of specialization in the relationships between plants and the facultatively entomopathogenic endophytes (cf. Gielen et al. [9], for an opposite example).

In addition to the highly similar overall prevalence of entomopathogenic fungi, another result consistent across comparable studies is the relatively high diversity of the entomopathogens attacking lepidopteran larvae. In the present study, eight species level taxa were represented among the 37 records. Similar diversity from Lepidoptera has been recorded in our Estonian [9,25] studies and from soils by some European (Italy [28] and Finland [29]) and Asian (China [30,31], Palestina [32]) works which used the “Galleria baiting” method. Notably, however, in contrast to all these studies, the Argentinean sample did not include any representatives of the family Cordycipitaceae Kreisel, known to comprise obligatory entomopathogens. It is tempting to speculate that such fungi cannot maintain permanent populations in urban areas, such as the one in which the present study was conducted, and/or do not occur as endophytes of plants used for feeding larvae in this study. There are numerous studies indicating that obligatory entomopathogens (especially Cordycipitaceae) tend to prefer soils with lesser anthropogenic influence [28,29,32]. To our knowledge there are, however, no studies aiming to compare aboveground communities of entomopathogens in this respect.

The numerically dominant families among the entomopathogens in Argentina—Aspergillaceae and Nectriaceae—were represented by the same genera but not by the same species in the fungal communities of folivorous lepidopterans in Estonia [9,25]. Yet both these assemblages included one, albeit different, member (SH1546416.08FU and SH2228332.08FU) of the *Fusarium solani* species complex, known to infect various lepidopteran hosts [7,33]. Comparable data from Asia [31] suggest that such level of divergence among distant areas (sharing genera but not species) may exemplify a global pattern in entomopathogenic fungi. This contrasts a global study of soil fungi which reports wider distribution of animal parasites compared to other ecological groups, suggesting a high number of species with global distribution [34]. However, distinctness of the composition in biogeographically distant communities is often observed for host-associated fungi, e.g., mushroom parasites [35,36]. Host identity is likely having a strong impact in structuring entomopathogenic communities, similar to what has been observed for endophytes [37]. The comparison of communities of entomopathogenic fungi across geographic locations has, however, been complicated by the scarcity of comparable data and the unresolved species-level taxonomy. In particular, phylogenetic studies have often shown that the names previously widely applied to various entomopathogenic species actually represent species complexes. To mitigate the problem, we have here adopted the UNITE SH system which allows assigning precise and persistent identifiers [23] to all members of observed communities (Table 2), also in the absence of a Latin binomial.

The abundance of the fungi from the families Aspergillaceae and Nectriaceae as pathogens of lepidopteran larvae, both in the present study and its counterparts in Estonia [9,25], is in some contrast with the tradition of not listing them among the main pathogens of insects, and Lepidoptera in particular (see, however, Santos et al. [5]). It is well possible that the reason for the somewhat unexpected high share of these families in our studies results from our focus on the previously largely overlooked larval stage. This indicates that further studies across different life stages of selected insect species are clearly warranted for comprehending the full diversity of entomopathogenic communities of fungi.

The treatment of various *Fusarium* species as opportunistic (rather than obligatory) pathogens [30] has emerged as a by-product of the 200 years of study focusing on mycotoxins produced by these fungi and their plant pathogenic tendencies. Such studies have been conducted mainly with strains from agricultural settings [5]. Accumulating data on *Fusarium* species found from studies on entomopathogens [5,7] have expanded our understanding on the lifestyles in this highly diverse fungal genus. The ability to infect insects appears to have evolved in several groups of *Fusarium*, characterized by different level of host specialization [5,33]. Our data fits into this pattern, as all *Fusarium* species found in our study belong to complexes previously known to harbor entomopathogenic strains [5]. A truly opportunistic lifestyle in some *Fusarium* spp. is in no way excluded, however, as the well-known opportunistic pathogens of humans [38] belong to *F. solani* and *F. oxysporum* but also to the *F. fujikuroi* and *F*. *incarnatum-equiseti* species complexes, all of these being represented as pathogens of lepidopteran larvae in this study.

Entomopathogenic properties of fungi from the family Aspergillaceae have received little attention. However, *A. flavus*, the second most abundant fungus found in this study, has been shown to infect insects from various orders [39,40], being especially effective in killing lepidopteran larvae [31,41]. In the light of accumulating data, to which the present study also complements, there seems to be a need for reevaluating the nutritional mode and lifestyle of several fungal taxa previously (mis)identified, as opportunistic pathogens [9,25,30]. Indeed, accumulating evidence reveals that, in certain species-rich ascomycete genera (*Fusarium*, *Penicillium*, etc.), an entomopathogenic lifestyle has repeatedly evolved in different lineages [5,15].

## 5. Conclusions

The current investigation on the prevalence of entomopathogenic fungi in central Argentina mirrored recent experiments in northern Europe. These studies explored the probability of lepidopteran larvae to gain infection as a function of insect species and the species of its food plant. The largely concordant results of these studies conducted over considerable geographical distance, while using taxonomically and ecologically distinct lepidopteran and plant species, enable us to outline general patterns in the prevalence of infection and the composition of fungal communities infecting immature stages of Lepidoptera. Lepidopteran species do not differ substantially in the probability to become infected, while larval food plant species have a considerable role in influencing the success of some fungal taxa, exemplified by a species from the *F. fujikuroi* complex in Argentina and *Akanthomyces muscarius* in Estonia. The relatively low fungal prevalence in immature stages of folivorous lepidopterans contrast with the rather high diversity in their fungal communities. The main part of that diversity, as well as the community dominants, belong to the ascomycete order Hypocreales, yet not only to the best-known family of entomopathogens, the Cordycipitaceae, but also to the genus *Fusarium* from the Nectriaceae. The fungal communities detected from Estonia and Córdoba Province in Argentina add evidence to the recent reconsideration of the nutritional modes in *Fusarium* in distinguishing the role of some species (complexes) in causing insect infections. Until recently, members of this genus were only considered opportunistic pathogens of insects (as well as of other animals, including humans), next to their widely acknowledged role as saprotrophs and plant pathogens. The overlap of genus-, but not species-level composition in the communities of fungal entomopathogens of the biogeographically distinct study areas conforms to a common pattern in the global distribution of fungi.

## Figures and Tables

**Figure 1 life-12-00974-f001:**
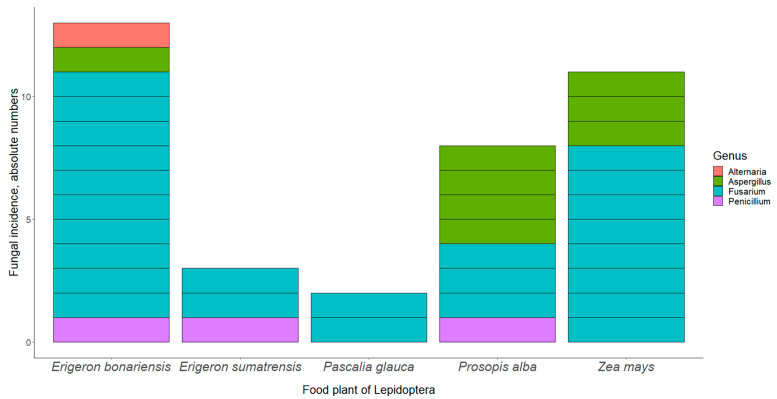
Fungal incidence according to food plant of the lepidopteran host. The number of larvae reared on *E. bonariensis* was 182, *E. sumatrensis* 125, *P. glauca* 134, *P. alba* 382, and *Z. mays* 155.

**Table 1 life-12-00974-t001:** Collected adult lepidopterans with the site and date of collection indicated, number of offspring larvae entering the experiment, and the number of larvae recorded as infected by a fungus.

Lepidopteran Species	Mother/Brood No	Collected From	Date	No of Larvae	Larvae Died of Fungi
*Anicla infecta*	AI4	31°18′22″ S 64°20′43″ W	4 January 2020	73	0
	AI6	31°08′21″ S 64°21′48″ W	9 January 2020	86	0
	AI7	31°08′21″ S 64°21′48″ W	9 January 2020	42	3
	AI8	31°08′21″ S 64°21′48″ W	9 January 2020	6	0
	AI9	31°08′21″ S 64°21′48″ W	9 January 2020	12	1
	AI16	31°18′22″ S 64°20′43″ W	28 January 2020	20	3
*Dargida albilinea*	FA17	31°51′26″ S 63°44′16″ W	17 December 2019	116	4
	FA20	31°51′26″ S 63°44′16″ W	17 December 2019	93	2
*Hypercompe indecisa*	HI1	31°18′22″ S 64°20′43″ W	23 December 2019	221	12
	HI3	31°18′22″ S 64°20′43″ W	4 January 2020	309	12

**Table 2 life-12-00974-t002:** Fungal species detected on lepidopteran hosts and their food plants. For all fungal species, we present codes of UNITE species hypothesis (SH) to which representative ITS rDNA sequences were assigned. Number of host insect individuals affected and the food plant of those are indicated for each fungal species.

Order/Family	Species	SH DOI	Lepidopteran Species	Food Plant
**Eurotiales**/Aspergillaceae	*Aspergillus flavus* Link	SH1532328.08FU	*A. infecta* 1 *D. albilinea* 2 *H. indecisa* 5	*E. bonariensis* 1 *P. alba* 4 *Z. mays* 3
	*Penicillium* sp. ^1^ Link		*A. infecta* 1 *H. indecisa* 2	*P. alba* 1 *E. bonariensis* 1 *E. sumatrensis* 1
**Hypocreales**/Nectriaceae	*Fusarium chlamydosporum* sc*	SH1610186.08FU	*D. albilinea* 1	*Z. mays* 1
	*Fusarium fujikuroi* sc*	SH1610157.08FU	*A. infecta* 3 *D. albilinea* 1 *H. indecisa* 8	*E. bonariensis* 10 *Z. mays* 2
	*Fusarium incarnatum-equiseti* sc*	SH1458596.08FU	*A. infecta* 1 *D. albilinea* 1 *H. indecisa* 7	*E. sumatrensis* 2 *P. alba* 3 *Z. mays* 4
	*Fusarium oxysporum* sc* ^2^	SH1656686.08FU	*D. albilinea* 1	*P. glauca* 1
	*Fusarium solani* sc*	SH1623679.08FU	*A. infecta* 1 *H. indecisa* 1	*P. glauca* 1 *Z. mays* 1
**Pleosporales**/Pleosporaceae	*Alternaria* sp. ^3^ Nees.	SH1526648.08FU	*H. indecisa* 1	*E. bonariensis* 1

^1^ Identified by morphological features. As we failed to obtain DNA sequences of satisfactory quality, we prefer to keep these fungi identified at the genus level. ^2^ The partial ITS sequence obtained revealed 100% similarity to sequences of *F. foetens*, including one from the holotype. ^3^ The same SH identified on *Hypomecis atomaria* L. from Estonia [9]. Sc*- species complex.

**Table 3 life-12-00974-t003:** The incidence of fungal infections (as a binary trait: yes/no) as dependent on lepidopteran species (the host for the fungi) and food plant species (nested in Lepidoptera species, abbreviated Lep. sp.) as analyzed by generalized linear models for binary data (car package of the R system [24]), type III analysis. The same analysis preformed separately for the most abundant fungal genus *Fusarium* and the most abundant species, a representative of the *Fusarium fujikuroi* species complex.

	df	χ^2^	*p*
Total data			
Lepidoptera sp.	2	1.8	0.41
**Food plant (Lep. sp.)**	**12**	**29.9**	**0.003**
Genus *Fusarium*			
Lepidoptera sp.	2	0.05	0.97
Food plant (Lep. sp.)	9	8.9	0.45
*Fusarium fujikuroi*			
Lepidoptera sp.	2	0.85	0.65
**Food plant (Lep. sp.)**	**9**	**26.1**	**0.002**

## Data Availability

Metadata for all fungi detected in this study have been uploaded to PlutoF and can, along with DNA sequences and images, be accessed at https://dx.doi.org/10.15156/BIO/2483906.

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
