# Peer review of "Entomopathogenic Fungi Infecting Lepidopteran Larvae: A Case from Central Argentina"

_life, 2022, doi:10.3390/life12070974_

Round 1

Reviewer 1 Report

Dear authors,

The paper is very interesting for the region studied as well as for the entire scientific community. Please technically rearrange the paper based on the instructions for authors requested by the journal.

Best regards!

Author Response

We thank the reviewer for kind response. However, we are not quite sure what does the reviewer mean by “technically rearrange the paper “? We have followed the instructions available for submitting authors at the journal’s webpage and would appreciate pointing to specific issues/places where rearrangement is needed.

Reviewer 2 Report

1.     It will be nice to provide results in abstract rather than general information.

2.     Why there was a delay in publication. Collection sounds in 2019-2020.

3.     Larvae had the leaves of indicated plants as there was no choice for them but what is the situation in real environment? Do the results of the study are appropriate to conclude for the open environment? Justify this.

4.     Provide images of larvae and plant species.

Author Response

  1. It will be nice to provide results in abstract rather than general information.

We thank the reviewer, it is true, that abstract should also provide major results, as is also the case in our manuscript (lines 24-30). We consider these to suffice but if the reviewer had some particular results in mind we would be ready to add such.

  1. Why there was a delay in publication. Collection sounds in 2019-2020.

The worldwide pandemic was the reason why the writing of this paper was slightly postponed. (Indeed, the first lockdown started just after completion of our fieldwork in Argentina). However, we are certain that this short time gap between collecting and publishing the data does in no way diminish its value or the quality of the paper.

  1. Larvae had the leaves of indicated plants as there was no choice for them but what is the situation in real environment? Do the results of the study are appropriate to conclude for the open environment? Justify this.

We thank the reviewer for this comment. It is true that we limited the diet of larvae to only one food plant. As stated by Braga and Janz (2021) & Singer and Wee (2005), in the abundance of food larvae regularly do not tend to change the food plant. We have incorporated an explanation to the material and methods section (lines 97-98).

Braga, M.P., Janz, N., 2021. Host repertoires and changing insect–plant interactions. Ecological Entomology 46, 1241–1253. https://doi.org/10.1111/een.13073

Singer, M.C., Wee, B., 2005. Spatial pattern in checkerspot butterfly—host plant association at local, metapopulation and regional scales. Annales Zoologici Fennici 42, 347–361.

4. Provide images of larvae and plant species.

Images and the metadata for specimens are provided at https://dx.doi.org/10.15156/BIO/2483906 as also stated in the manuscript (lines 145-146 & 291-293)

Reviewer 3 Report

Manuscript by Robin et al is a nice piece of work. Work will be of importance for entomologist/plant or crop protection/plant pathology people. Although I do not have anything about the scientific content of study but I think author should address there minor issue so that manuscript can be accepeted for publication. My specific comments to author are

1) Need minor english or writing improvement throughout the draft.

2) need to discuss more details in experimental section.

3) Author should also mention whether any chemical spray was done on plants whose leafs were used as feed by insects.

4) There is no error bars in histogram (need to look into this or explain why it is missing).

5) Should also mention what platform was used for sequencing purpose, coverage

6) If possible should also include image of plates showing fungal colonies (mentioned in draft)

7) Since date of collection was mostly in Dec-Jan, author should mention reasons about this, whether it is due to crop plants or due to insect life cycle or both

Author Response

1.Need minor english or writing improvement throughout the draft.

We thank the reviewer for kind words and have looked through the manuscript. Proofreading was done by a native speaker before submitting, however, we would be happy to incorporate changes the reviewer had in mind considering the use of language.

2. need to discuss more details in experimental section.

We have incorporated all relevant information regarding the experiment and the analyses but would be happy to comply if the reviewer would elaborate what exactly is missing from the material & methods section.

3. Author should also mention whether any chemical spray was done on plants whose leafs were used as feed by insects.

We thank the reviewer for pointing this out and have incorporated the info that no chemical deterrents were used in the experiment into the manuscript (line 109).

 4. There is no error bars in histogram (need to look into this or explain why it is missing).

Our manuscript did not include histograms. We guess that the figure in question is figure 1. In that case, as the figure is indicating counts of incidence as absolute numbers (as stated also on y axis), there is no measuring error to be counted for.

 5. Should also mention what platform was used for sequencing purpose, coverage

We thank the reviewer for pointing out this shortcoming and have written it out in our manuscript (lines 119-120)

 6. If possible should also include image of plates showing fungal colonies (mentioned in draft)

Images linked to the specimens are provided at https://dx.doi.org/10.15156/BIO/2483906 as also stated in the manuscript (lines 145-146 & 291-293)

7. Since date of collection was mostly in Dec-Jan, author should mention reasons about this, whether it is due to crop plants or due to insect life cycle or both

This was due to Dec-Jan being the mid-summer at the latitude in the Southern Hemisphere where our experiment was conducted in Argentina. This is the best time for such an experiment due to the abundance of plants and the life stage of insects, i.e. adults laying eggs and larvae developing.

Round 2

Reviewer 2 Report

Authors revised the manuscript extensively, therefore I recommend acceptance in its current form.